

# Development and Validation of a Dense 18-Year Time Series of Surface Water Fraction Estimates from MODIS for the Mediterranean Region

Linlin Li[1], Andrew Skidmore[1, 2], Anton Vrieling[1], Tiejun Wang[1]

[1]Faculty of Geo-information Science and Earth Observation, University of Twente, Enschede, The Netherlands.
[2]School of Environmental Sciences, Macquarie University, NSW, Australia.

*Correspondence to:* Linlin Li (l.li-1@utwente.nl)

**Abstract.** Detailed knowledge on surface water distribution and its changes is of high importance for water management and biodiversity conservation. Landsat-based assessments of surface water, such as the Global Surface Water (GSW) dataset developed by the European Commission Joint Research Centre (JRC), may not capture important changes in surface water during months with considerable cloud cover. This results in large temporal gaps in the Landsat record that prevent accurate assessment of surface water dynamics. Here we show that the frequent global acquisitions by the Moderate Resolution Imaging Spectrometer (MODIS) sensors can compensate for this shortcoming, and in addition allow for examining surface water changes at fine temporal resolution. To account for water bodies smaller than a MODIS cell, we developed a global rule-based regression model for estimating surface water fraction from a 500 m nadir reflectance product from MODIS (MCD43A4). The model was trained and evaluated with the GSW monthly water history datasets. A high estimation accuracy ($R^2 = 0.91$, RMSE = 11.41%, MAE = 6.39%) was achieved. We then applied the algorithm to 18 years of MODIS data (2000-2017) to generate a time series of surface water fraction maps at 8-day interval for the Mediterranean. From these maps we derived metrics including the mean annual maximum, standard deviation and seasonality of surface water. The dynamic surface water extent estimates from MODIS were compared with the results from GSW and water level data measured in situ or by satellite altimetry, yielding similar temporal patterns. Our datasets complement surface water products at fine spatial resolution by adding more temporal detail, which permits effective monitoring and assessment of the seasonal, inter-annual and long-term variability of water resources, inclusive of small water bodies.

## 1 Introduction

Terrestrial surface water bodies such as lakes, reservoirs, and rivers cover approximately 3% of the global land mass. They play a crucial role in the global hydrological cycle, biodiversity conservation and climate process (Tranvik et al., 2009;Chahine, 1992). Detailed knowledge on surface water distribution, and its seasonal, inter-annual and long-term variability can serve as an important source for water management (Cole et al., 2007), ecosystem assessment and biodiversity conservation (Turak et al., 2017). Remote sensing data has increasingly been used to monitor surface water changes, and powerful methods and tools





have been developed for analyzing Earth observation data. However, existing approaches for monitoring the surface water extent are limited either in geographic scope, temporal extent of the record, or in temporal frequency of observations.

At the global scale, several static datasets exist that provide information on the spatial extent of water bodies and wetlands. For example, the Global Lakes and Wetlands Dataset (GLWD: Lehner and Doll, 2004) was based on historical maps and has

a spatial resolution of 30 arc-seconds (approx. 1 km). Carroll et al. (2009) combined the Shuttle Radar Topography Missions (SRTM) Water Body Dataset (SWBD) with 250 m Moderate Resolution Imaging Spectrometer (MODIS) reflectance data to produce a global static map of surface water for circa 2000-2002. Global Landsat-based static surface water datasets include the 3 arc-second (~90 m) Water Body Map (G3WBM: Yamazaki et al., 2015) and the Global Land Cover Facility (GLCF) inland surface water dataset at 30 m resolution for 2000 (Feng et al., 2015).

Even though static water maps are adequate for some applications there is an increasing demand for information on the spatio-temporal variability of inland water bodies and their long-term evolution (Belward, 2016). Dynamic mapping and monitoring surface water extent have been explored using optical sensors featuring fine (10-30 m) to medium (250-500 m) spatial resolution. At fine spatial resolution, several studies have recently presented interesting results on long-term variability of surface water with the entire Landsat archive at regional (Halabisky et al., 2016;Heimhuber et al., 2016), continental (Mueller

et al., 2016), and global scale (Pekel et al., 2016;Donchyts et al., 2016). The European Commission Joint Research Centre's (JRC) global surface water (GSW) datasets (Pekel et al., 2016) quantified changes in global surface water over the past 32 years with a monthly time interval. This product allows the analysis of surface water dynamics over long time periods at fine spatial resolution, but only provides information on monthly changes in surface water. Moreover, the Landsat archive also contains data gaps and temporal discontinuities depending on geographical location (Pekel et al., 2016). This is due to both

the limited number of acquisitions during specific time intervals, and due to location- and time-dependent persistency of cloud cover. These data gaps affect the accuracy of the seasonality information (Yamazaki and Trigg, 2016). To better represent water bodies with short hydroperiods and short-duration flooding, it is critical to account for such gaps when monitoring surface water. In recent years, the revisit time of fine-resolution sensors has increased (e.g., Sentinel-2 offers a 5-day repeat since March 2017: Du et al., 2016). However, these data cannot yet be used to create long-term (>10 year) consistent time

series at short time intervals.

Moderate resolution imagery derived from satellite sensors such as MODIS provides daily observations over long time-spans and as such has potential to construct long-term and dense time series of surface water over large regions. Many studies have explored the use of MODIS in mapping water body dynamics at regional to continental scale (Kaptue et al., 2013;Sharma et al., 2015;Pekel et al., 2014) using binary classification methods. At global scale, Khandelwal et al. (2017) use MODIS

multispectral data to map the global extent and temporal variations of 94 large reservoirs at 500 m resolution and at a nominal 8-day interval from 2000 to 2015. The recent Global Climate Observing System (GCOS) report states that Essential Climate Variables (ECV) need to be established for water extent and lake ice cover products ideally with daily temporal resolution (Belward, 2016). To address this requirement, the first daily global dataset of inland water bodies at 250 m spatial resolution from 2013 to 2015 was developed by Klein et al. (2017). This work advanced surface water mapping using remote sensing



regarding to its dense temporal resolution, and enhanced our understanding of rapid water changes caused by extreme climate change and human activities. However, like other surface water mapping efforts based on binary classification methods (e.g., Khandelwal et al., 2017;Mohammadi et al., 2017), this product misses out on lakes and narrow rivers that only cover a portion of a MODIS resolution cell.

To overcome this limitation and incorporate small water bodies, several researchers have attempted to predict sub-pixel surface water estimates of MODIS by providing water fraction in each pixel using techniques like linear spectral mixture modeling (e.g. Li et al., 2013;Olthof et al., 2015;Hope et al., 1999) and machine learning (e.g., Rover et al., 2010;Sun et al., 2012;Li et al., 2018) for small areas. However, the utility and efficiency of these methods have rarely been explored for the estimation of surface water fraction for larger areas. In our previous work (Li et al., 2018), we explored the use of rule-based regression

models over two small areas on the Iberian peninsula and concluded that a single global regression model can provide accurate surface water estimates across areas with different environmental conditions as long as it is fed with training data that comprise these various conditions. Consequently, we concluded that this approach has potential to be applied over much larger areas. Therefore, the aim of this paper is to explore the utility and efficiency of rule-based regression model for the estimation of surface water fraction for the Mediterranean region, and to develop a new surface water fraction dataset for the Mediterranean

region using fine temporal resolution MODIS data as input for the effective assessment of seasonal, intra-annual and long-term surface water dynamics inclusive of small water bodies. Our specific objectives are:

(1) to develop an approach for the estimation of surface water fraction for the Mediterranean region at fine temporal resolution from MODIS data;

(2) to generate an 8-day interval time series of surface water fraction maps for the Mediterranean from 2000 to 2017, and to

use that to derive a series of ecologically-relevant metrics;

(3) to compare our dataset with an existing dataset (i.e. JRC's GSW) and water level data to assess how they compare in space and time.

## 2 Study area

We loosely defined the Mediterranean in this study as the region that is comprised within ten MODIS grid tiles, which together

cover all coastal areas of the Mediterranean and Black Sea, including a significant portion of their inland areas (Fig.1). This boundary is defined based on the combination of (1) the definition of the Mediterranean region by the Mediterranean Wetland Observatory (MWO) project, i.e. 27 Mediterranean countries are included by MWO; (2) the inclusion of areas with a large amount of Ramsar wetlands; (3) the exclusion of southern parts of north African countries (i.e. Morocco, Algeria, Libya and Egypt) that comprise few water bodies according to the maximum water extent over 32 years from JRC's GSW product.

The study region covers 13 climate zones as defined by Peel et al. (2007) (Fig. 1). Numerous water bodies of different types are found in this region, including large coastal lagoons, fresh, brackish or salt marshes, riverine forests and reed beds, flood plains and wet meadows, mountainous lakes and surrounding wetlands, salted lakes, temporary marshes and streams (Costa et



al., 1996). Our study area accounts for twenty-five percent of the world's Ramsar sites that contain a great ecological, social and economic value, especially as they provide habitat, reproduction and migration stopover sites for numerous bird species (MWO, 2012). A good number of the water bodies and wetlands in the region are small, shallow and highly variable between seasons and years due to weather effects and human activities (Costa et al., 1996).

5 Many Mediterranean water resources are degraded mainly due to urbanization, agricultural reclamation, increasing water use for irrigation, and hydraulic works such as dams, dikes, river channeling, and drainage and irrigation networks (Batalla et al., 2004). A number of projects and programs performed monitoring of surface water and wetlands in the Mediterranean region, such as MWO (http://medwet.org/) and the GlobWetland initiative (http://www.globwetland.org/), which highlighted the importance of protecting Mediterranean water resources. However, these projects either performed wetland mapping for a few

10 moments in time (e.g., GlobWetland only covered 1975, 1990 and 2005), or are limited to specific water bodies and wetlands instead of the whole landscape. Although surface water dynamics in the Mediterranean can be analyzed at fine spatial resolution with JRC's GSW, it has large spatial and temporal gaps. Fig. 2 shows that no valid observation exists for December of the years 2000-2015 in Mediterranean areas according to the GSW monthly water history map. In addition, less than 10% of the area has observations for January.

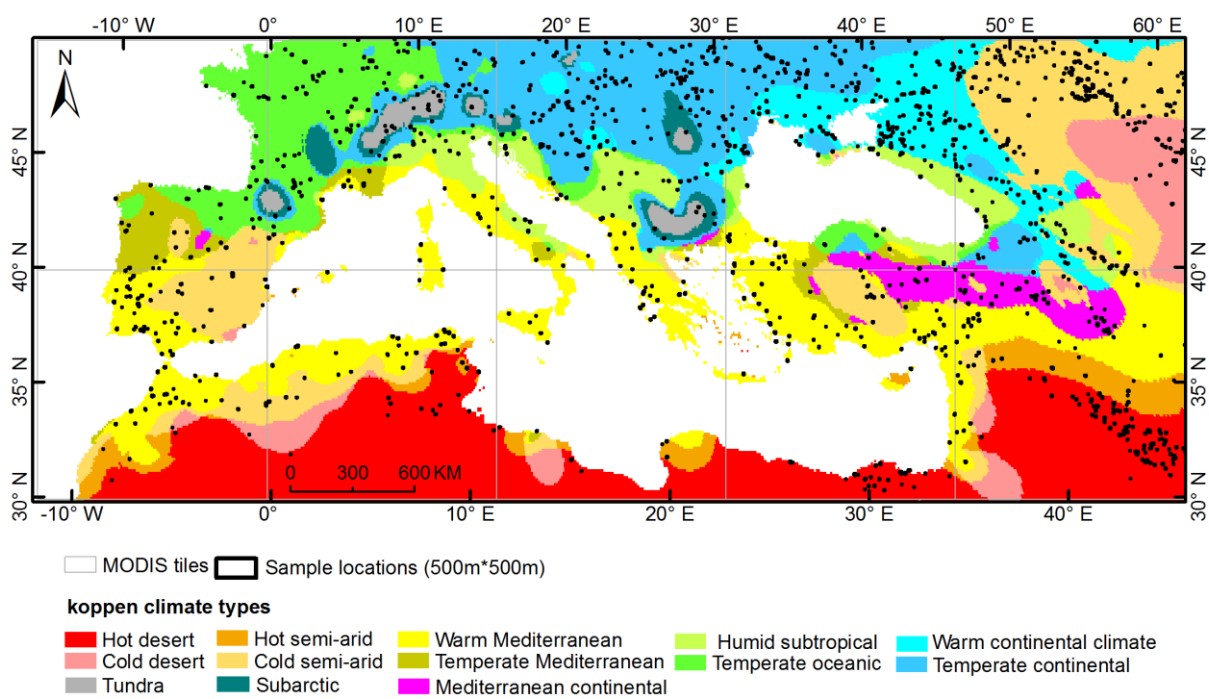

Figure1 Study area and sample locations





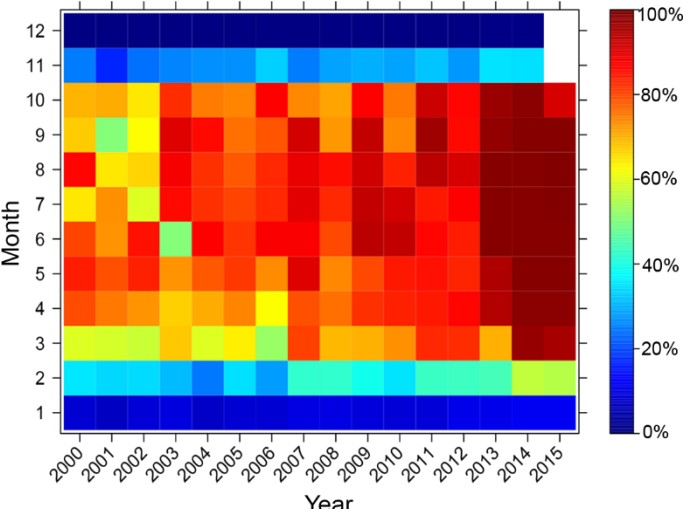

Figure 2 Percentage of valid data points in JRC's Global Surface Water (GSW) monthly water history dataset for each

month between January 2000 and October 2015, taken as a spatial average for the Mediterranean region as displayed in

Figure 1

5    **3 Data**

Table 1 summarizes all data sets used in this study. They include a number of sources used to derive model input variables,

training data for building the model, validation data for model accuracy assessment, and other existing surface water products

against which we compared our products. Details are provided in the following sections.

10                                              Table 1 Input and reference datasets used in the study

| Data/product name | Temporal resolution | Spatial resolution | Purpose in the study |
|---|---|---|---|
| MCD43A4, V006 (MODIS/Terra and Aqua Nadir BRDF-Adjusted Reflectance) | daily & 8-day | 500 m | Generation of predictor variables and production of time series of water fraction maps |
| MCD43A2, V006 (MODIS/Terra and Aqua BRDF/Albedo Quality) | daily | 500 m | Snow and ice mask |
| GSW monthly water history datasets | monthly | 30 m | Generation of training and validation datasets, and thematic products |
| GSW maximum water extent map | static | 30 m | Define sampling strata; Exclusion of non-water samples from training locations |
| GSW water transitions map | static | 30 m | Define sampling strata |



| Digital elevation model from Shuttle Radar Topography Mission (SRTM) 3 v4.1 | static | 90 m | Generation of predictor variables; Identification of sloping terrain and terrain shadows |
|---|---|---|---|
| USGS Landsat archive | 16-day | 30 m | Link the GSW monthly history datasets to a single date of cloud-free Landsat acquisition because the exact date of observation is not included in the GSW dataset |
| MCD12Q1 (MODIS Land Cover Type product) | static | 500 m | Identification of building shadows |
| Land Water Mask derived from MODIS and SRTM (MOD44W) | static | 250 m | Comparison of products |
| Water level from satellite altimetry | 10-day | - | Validation of results |
| Water level from *in situ* | daily | - | Validation of results |

### 3.1 MODIS Data

The main input dataset in this study is the MODIS Terra and Aqua Nadir BRDF-Adjusted Reflectance (NBAR) product (MCD43A4, V006). This product provides 500 m resolution surface reflectance data for each of the MODIS bands (1-7) corrected to a common nadir view geometry at the local solar noon zenith angle using a bi-directional reflectance distribution
function (BRDF) model (Schaaf and Wang, 2015a). Compared to the previous collection (V005) that had an 8-day frequency, the collection V006 was retrieved on a daily basis. Each daily value is a result of compositing information obtained during 16 days of observations, which are weighted as a function of the quality, the observation coverage and the temporal distance from the day of interest. Each daily V006 retrieval is the center (i.e. the 9th day) of the moving 16 day input window (Schaaf and Wang, 2015a).

We also used the MCD43A2 (V006) Bidirectional Reflectance Distribution Function and Albedo (BRDF/Albedo) Quality dataset to filter out pixels with snow and ice in the MCD43A4 product. This dataset has the same temporal and spatial resolution as MCD43A4 (i.e. daily 500 m resolution), and contains quality information for the corresponding MCD43A4 NBAR product including snow and ice presence (Schaaf and Wang, 2015b).

For this study, we downloaded the daily files of MCD43A4 and MCD43A2 for 2000, 2003, 2006, 2009, 2012 and 2015. As
explained in Section 4.1, all the available dates (or months) of the GSW monthly water history datasets in these six years over the sample locations were used for building training and validation data. Instead, when producing the surface water fraction time series, we collected the MCD43A4 and MCD43A2 files by an 8-day time step form February 2000 to December 2017, because processing daily files for the large study area and the 18-year time period would become too time- and memory-consuming. The 8-day repeat coverage is considered to be a minimum for effectively capturing water bodies with short
hydroperiods while simultaneously accounting for frequent cloud cover (Wulder et al., 2016;Guerschmann et al., 2011). All images were downloaded through the NASA Earthdata Search website (https://search.earthdata.nasa.gov/search).



## 3.2 Global Surface Water (GSW) dataset

To generate training and validation data for modeling surface water fraction, we used the GSW dataset (Pekel et al., 2016). This dataset provides global distribution of surface water extent at monthly time interval from March 1984 to October 2015 (380 months) at 30 m spatial resolution, and also includes a series of thematic maps summarizing different facets of the spatial and temporal dynamics of surface water over 32 years. This dataset is derived from the entire archive of Landsat 5 Thematic Mapper (TM), the Landsat 7 Enhanced Thematic Mapper-plus (ETM+) and the Landsat 8 Operational Land Imager (OLI). Water detection was performed using a dedicated expert system, which is a procedural sequential decision tree used both the multispectral and multitemporal attributes of the Landsat archive as well as ancillary data layers (Pekel et al., 2016). Based on a validation with very high resolution satellite and aerial imagery, the authors reported a high mapping accuracy with a commission accuracy of 99.45% and an omission accuracy of 97.01% (Pekel et al., 2016).

The GSW monthly water history datasets are available in Google Earth Engine (GEE) as an image collection containing 380 images, one for each month between March 1984 and October 2015. Each image provides a binary classification of water presence, or indicates if for a specific pixel and month no valid (cloud-free) Landsat observations were available. For comparison with MODIS data, we used the monthly water history datasets between February 2000 and October 2015, which results in 189 images.

Several GSW thematic maps were derived from the GSW monthly water history datasets (Pekel et al., 2016). In this study, we used two thematic maps, i.e. the maximum water extent map and water transitions map, for the Mediterranean region from the data access website (https://global-surface-water.appspot.com/download). The maximum water extent map indicates for each 30m grid cell if ever detected as water over the 32-year period. The transition map contains 10 water classes (i.e. permanent, new permanent, lost permanent, seasonal, new seasonal, lost seasonal, seasonal to permanent, permanent to seasonal, ephemeral permanent and ephemeral seasonal). It provides information on both intra- and inter-annual variability of surface water (Pekel et al., 2016).

## 3.3 Terrain data

Terrain data is useful for predicting locations for water bodies (Drake et al., 2015;Grabs et al., 2009). In this study, we used the near-global Shuttle Radar Topography Mission (SRTM) Digital Elevation Model (DEM) distributed by the Consortium for Spatial Information of the Consultative Group of International Agricultural Research (CGIAR-CSI). This product has ~90 m resolution and is a post-processed derivative to address areas of missing data in the original SRTM DEM made by the National Aeronautics and Space Administration (Jarvis et al., 2008). The most recent version of this product is SRTM3 v4.1 and is freely available from http://srtm.csi.cgiar.org/.



### 3.4 Satellite altimetry and in situ water level

We obtained water levels from the U.S. Department of Agriculture Global Reservoir and Lake Monitoring (GRLM) website (http://www.pecad.fas.usda.gov/cropexplorer/global_reservoir). This site provides time-series of water level variations for some of the world's largest lakes and reservoirs mainly greater than 100 km$^2$. The GRLM utilizes near-real time data from the

Jason-3 mission, and archive data from the Jason-2/OSTM, Jason-1, Topex/Poseidon, and ENVISAT satellites. We also obtained daily in situ gauge observations for Fuente de Piedra Natural Reserve in southern Spain, which were also used in Li et al. (2015).

### 3.5 Additional data

We utilized the MODIS land cover type product (i.e. MCD12Q1: Friedl et al., 2010) to identify and mask areas that potentially

have commission errors related to building shadows. To assess the spatial accuracy of MODIS derived maps, we also used the land water mask derived from MODIS 250 m and SRTM data (MOD44W: Salomon et al., 2004) for comparison.

### 4 Method

### 4.1 Approach for deriving surface water fraction

The approach used to derived surface water fraction builds on our previous work (Li et al., 2018) with considerable

improvements regarding input data, training data and commission error processing. We explored the use of MODIS spectral information and a topographic metric for estimating surface water fraction over two study areas in Spain through the use of rule-based regression models and concluded that a single global regression model can be effectively tuned locally as long as it is fed with training data that comprise the various environmental conditions encountered across the larger area (Li et al., 2018). In this sense, the approach for constructing a global model can be expanded effectively to wider areas such as the

Mediterranean region. The following sub-sections and Fig. 3 describe the individual steps of the approach in detail.



Figure 3 Diagram of our approach for deriving surface water fraction

### 4.1.1 Selection of sample locations for training and validation

Sample locations were selected using a two-stage stratified random sampling method. First, a total of 13 strata were defined based on climate zones (see Fig. 1). We created sampling blocks by partitioning the study area into ~5x5 km grids (i.e. 10x10 MODIS pixels as one block) and assigned each block to the climate zone within its spatial footprint. Blocks that cover more than one climate zone and those that contain no surface water based on JRC maximum water extent product were excluded from our sample. We then selected 1400 blocks (~2% of all resulting blocks) using stratified random sampling.



Second, we created 500x500 m grids corresponding to the MODIS geometry in each of the 1400 blocks. Fourteen strata were defined based on the combination of water fraction categories and water permanence types. Specifically, we first divided all grid cells into seven water fraction categories (0%, 0%–20%, 20%–40%, 40%–60%, 60%–80%, 80%–100% and 100%) according to the aggregated GSW maximum water extent. Then for each category, we further classified water permanence types based on the aggregated GSW water transitions map. For 20%–40%, 40%–60%, 60%–80%, 80%–100% or 100% category, grid cells were further classified as fluctuating water if they contain more than 20% fluctuation water otherwise they were assigned as permanent water. For 0%–20% category, grids with 0% permanent water were classified as fluctuating water while grids with 0% fluctuation water were classified as permanent water. The rest of the grids in 0%–20% category were not assigned because of very low water fraction. In the end, the 14 strata were: 100% permanent, 100% fluctuating, 80%–100% permanent, 80%–100% fluctuating, 60%–80% permanent, 60%–80% fluctuating, 40%–60% permanent, 40%–60% fluctuating, 20%–40% permanent, 20%–40% fluctuating, 0%–20% permanent, 0%–20% fluctuating, 0%–20% no water class, and 0% no water class. From each of the 14 strata, we randomly selected 500 grid cells. This resulted in a set of 7000 MODIS-scale reference grid cells (shown in Fig. 1), which were further split into 3500 training and 3500 validation locations using random sampling from each strata.

Sampling times were selected at a constant interval of three years (i.e., 2000, 2003, 2006, 2009, 2012 and 2015). All available dates/months in the GSW monthly history datasets from those selected years were used.

### 4.1.2 Building training and validation datasets

The GSW monthly water history maps were used for generating training and validation data. Specifically, the 30 m monthly water history maps from all sampling years/month were aggregated to the 500 m resolution for all sample locations in GEE by

dividing the 30 m water pixels with the total number of 30 m pixels within each 500 m resolution cell, resulting in a surface water fraction that we used as a reference. Given that the exact dates of these monthly water history maps are not provided with the GSW product, we linked these reference estimates to the USGS Landsat archive that GSW used as its input. For each combination of location/month, we retained only those reference estimates for which the location was covered by a single Landsat tile acquired during that month. If multiple Landsat tiles existed in that month for that location, we only retained the

reference estimates if all but one Landsat tile had 100% cloud cover. In this way, we could accurately assign a precise date to the retained reference estimates.

Surface water fraction estimates derived from all months of the sample years for training locations were used as the training dataset, and the estimates from all sample months for validation locations were used as the validation dataset.

### 4.1.3 Modelling surface water fraction and accuracy assessment

Surface water fraction was estimated using MODIS spectral information and derived water indices, and a topographic metric through a rule-based regression model. All predictor variables (Table 2) evaluated by Li et al. (2018) were used as input for the estimation of surface water fraction. In addition, the annual mean, minimum, maximum, standard deviation, and coefficient





of variation (CV) of each MODIS-derived predictor variable were included as input in the model for predicting surface water fraction.

Table 2 Overview of the predictor variables used in this study. MODIS shortwave infrared bands are referred to as $SWIR_1$ (1230-1250 nm), $SWIR_2$ (1628-1652 nm) and $SWIR_3$ (2105-2155 nm)

| Predictor variable | Formula | Reference |
|---|---|---|
| MODIS individual bands (Red, NIR, Blue, Green, $SWIR_1$, $SWIR_2$, $SWIR_3$) | - | - |
| NDVI-Normalized Difference Vegetation Index | (NIR-Red)/(NIR+Red) | Tucker (1979) |
| NDWI-Normalized Difference Water Index | (Green-NIR)/(Green +NIR) | McFeeters (1996) |
| MNDWI-Modified Normalized Difference Water Index | (Green-$SWIR_2$)/(Green+$SWIR_2$) | Xu (2006) |
| NDWI-Normalized difference water index (referred as $LSWI_{B5}$) | (NIR-$SWIR_1$)/(NIR+$SWIR_1$) | Gao (1996) |
| LSWI- Land surface water index (referred as $LSWI_{B6}$) | (NIR-$SWIR_2$)/(NIR+$SWIR_2$) | Xiao et al. (2002) |
| TCWI-Tasseled Cap Wetness Index | 0.10839*Red+0.0912*NIR+0.5065*Blue+0.404*Green-0.241*$SWIR_1$-0.4658*$SWIR_2$-0.5306*$SWIR_3$ | Zhang et al. (2002) |
| TCBI-Tasseled Cap Brightness Index | 0.3956*Red+0.4718*NIR +0.3354*Blue+0.3834*Green+0.3946* $SWIR_1$+0.3434*$SWIR_2$+0.2964*$SWIR_3$ | Zhang et al. (2002) |
| Value (HSV) | max($SWIR_2$, NIR, Red) | Pekel et al. (2014) |
| Saturation (HSV) | 1-min($SWIR_2$, NIR, Red)/max($SWIR_2$, NIR, Red) | Pekel et al. (2014) |
| Hue (HSV) | $0$, if $V = \min(SWIR_2, NIR, Red)$ <br> $\left(\dfrac{60° \times (NIR - Red)}{V - \min(SWIR_2, NIR, Red)} + 360°\right) \bmod 360°$, if $V = SWIR_2$ <br> $\dfrac{60° \times (Red - SWIR_2)}{V - \min(SWIR_2, NIR, Red)} + 120°$, if $V = NIR$ <br> $\dfrac{60° \times (SWIR_2 - NIR)}{V - \min(SWIR_2, NIR, Red)} + 240°$, if $V = Red$ | Pekel et al. (2014) |
| TWI-Topographic Wetness index | $\ln(\alpha/\tan\beta)$; $\alpha$ is upslope area per unit contour length (m) which calculated as (flow accumulation + 1) × (cell size); $\beta$ is the slope expressed in radians | Beven and Kirkby (1979) |

Cubist regression models (Quinlan, 1993) contain a set of conditional rules that partition the data space into smaller regions, each of which is linked to a multivariate linear regression model that can predict the explanatory variable (here surface water



fraction). Following the findings of our earlier work (Li et al, 2018), we used a single global Cubist regression model, but trained it with data collected from across the study area to tune the model to local conditions. In the global Cubist regression model, two parameters can be defined to optimize accuracy and reduce instability of the model prediction. The first is called "committees" indicating that multiple model trees are developed in sequence. Each member of the committee predicts the

target value and the members' predictions are averaged to give a final prediction (Quinlan, 1993). The second parameter is "neighbors" and allows the Cubist model to group similar samples in terms of predictor variable values, and determine the average prediction of these training samples (Quinlan, 1993;Kuhn et al., 2012). We tuned the models over different values of "committees" and "neighbors" ("committees" was set to be 0, 10 , 20, 50, 100, and "neighbors" was set to be 0, 1, 5, 9) through a 10-fold cross-validation on our training data and selected the values that produced the smallest root mean square error

(RMSE).

The resulting model was evaluated on both the training data that were used to generate the model and the independent validation data (Section 4.1.2). Three statistical measures were used to assess model performance: the coefficient of determination ($R^2$), mean absolute error (MAE) and RMSE.

### 4.1.4 Mapping and post processing

We applied the resulting model to the MCD43A4 V006 data from February 2000 to December 2017 to produce gridded time series of surface water fraction for the Mediterranean region with an 8-day time step resulting in 46 maps per year (except for 2000 that only contained 39 images).

Recent studies on surface water detection using optical sensors showed that multiple-sources of commission errors exist, such as terrain and building shadows (Pekel et al., 2016;Klein et al., 2017). These errors can be accounted for by using masks

derived from auxiliary data (Pekel et al., 2016;Klein et al., 2017). In this study we addressed two sources of commission errors: shadows from buildings and identified surface water presence that is unlikely on sloping terrain. Specifically, a slope map was derived from the SRTM DEM by calculating the maximum rate of change in elevation from each raster cell to their eight neighbors. Then we used a threshold of 5º to identify steep locations for which it is unlikely to find surface water but could have been detected as such by our model, for example due to spectral confusion between water and terrain shadows (Yamazaki

et al., 2015). In the case of building shadows, we used the urban class of the MODIS classification product MCD12Q1 (Friedl et al., 2010) to assign areas potentially affected by building induced shadows. Pixels were re-assigned to the 0% water fraction for slopes steeper than 5 ° and for areas classified as urban in MCD12Q1 except for those places where water is present according to GSW maximum water extent.

### 4.2 Generation of surface water fraction metrics and comparison of products

Based on gridded time series of surface water fraction, we derived a series of ecologically-relevant metrics that capture both the intra- and inter-annual variability and changes, and further compared these metrics with GSW-derived thematic products. The readily-available GSW thematic products were derived from 32 years of data (between March 1984 and October 2015),



thus cannot be directly compared against our MODIS-based results for 2000-2017. Therefore we reproduced the GSW thematic maps using the GSW monthly water history datasets from the overlapping period of these two datasets (i.e., February 2000 - October 2015). We then also computed the MODIS-derived temporal metrics for this period. In total, 189 GSW water history images and 720 surface water fraction maps were incorporated for the generation of these thematic maps. The following

metrics were generated:

(1) annual maximum and mean annual maximum surface water fraction between 2000 and 2015: the GSW monthly water history maps were summarized for each year in GEE to calculate the annual maximum surface water extent. We then aggregated the results of each year to MODIS resolution and averaged all years to derive the mean annual maximum surface water fraction. To assess the spatial agreement of the mean annual maximum surface water fraction derived from MODIS and

GSW, we calculated the surface water area (km$^2$) from the mean annual maximum surface water fraction using a threshold continuum. Specifically, the surface water fraction was partitioned using nine threshold values set in 10% increments from 0% to 100%. All pixels with surface water fraction greater than or equal to the threshold were summed and then multiplied by the MODIS pixel size. We then compared the water area (km$^2$) derived from the mean annual maximum surface water fraction based on GSW and MODIS across different continuum of threshold values. We also compared the water area with the 250 m

static water mask from MOD44W;

(2) Standard deviation of the annual maximum: a measure for the inter-annual variability of water presence;

(3) Seasonality: a measure for the seasonal and intra-annual variability of water presence. We calculated the number of times (i.e. the water occurrence) a given pixel has standing water above a certain water fractional threshold for a single hydrological year (October 2014 to September 2015) as GSW has a relatively large number of valid observations for this year (Fig. 2), and

then classified different types of water permanence. The water occurrence was calculated for each grid cell as a fraction of the number of times water was present relative to the total valid observations (i.e. not affected by clouds). We adopted the classification criteria of Guerschmann et al. (2011) and further modified it to be applicable to the Mediterranean region (Table 3). As the classes are not mutually exclusive, they are prioritized in the order shown in Table 3.

Table 3 Seasonality classification criteria based on surface water fractions and occurrence

| Class name | Surface water fraction | Water occurrence |
|---|---|---|
| Permanent water | ≥70% | ≥90% |
| Semi-permanent water | ≥70% | 70%-90% |
| Intermittent water | ≥70% | 20%-70% |
| Infrequent inundation | ≥70% | 1%-20% |
| Mixed permanent and semi-permanent water | 30%-70% | ≥70% |
| Mixed intermittent water | 30%-70% | 20%-70% |
| Mixed infrequent inundation | 30%-70% | 1%-20% |
| Never inundated | <30% | ≥0% |



## 4.3 Validation of MODIS temporal dynamics of surface water fraction

To assess the performance of our MODIS-derived product for monitoring temporal variations in surface water extent, we selected three lakes with fluctuating water presence. These three sites have varying sizes, different geographic locations and temporal dynamics. They have been listed in the International Conventions on Wetlands (known as Ramsar) given their importance for staging and wintering waterfowl.

(1) Fuente de Piedra, Spain: a shallow and saline lake with a maximum area of 13.6 km$^2$. It experiences strong seasonal, inter- and intra-annual variations of water level and inundation extent (Li et al., 2015);

(2) Lake Sabkhat al-Jabbul, Syria: a large, permanent saline lake that is surrounded by semi-arid steppe. At high water levels, it contains two islands. It traditionally floods in the spring and shrinks back during the summer and autumn but seldom dries out completely (JAES-CC, 2010);

(3) Doñana, Spain: a vast coastal marshland complex, separated from the ocean by an extensive dune system and subject to seasonal and inter-annual variations in water level (De Castro and Reinoso, 1997);

For the first two lakes, we compared the time series of surface water area derived from MODIS with that from GSW, and further compared these against water level data from satellite altimetry or in situ measurements. For Doñana, we compared the monthly spatial distribution of surface water extent derived from MODIS and JRC's GSW. To ensure the accuracy of the area calculations, we only calculated an area for times when both MODIS and GSW had at least 95% of valid cloud-free observations.

## 5 Results

### 5.1 Model performance

Following the model tuning of the Cubist regression model (see Section 4.1.3), we found that a 20-member committee and 9-neighbor model resulted in the smallest RMSE between actual (GSW-derived) water fraction and our MODIS-based water fraction estimates. The addition of more committees or neighbors had little effect on the accuracy.

Table 4 Statistical measures between predicted surface water fraction and actual data for training and validation data, and for different types of water permanence using validation data

|  | $R^2$ | RMSE (%) | MAE (%) |
|---|---|---|---|
| Training data | 0.93 | 9.79 | 5.61 |
| Validation data | 0.91 | 11.41 | 6.39 |
| Permanent water | 0.90 | 12.07 | 6.38 |
| Fluctuating water | 0.85 | 12.60 | 8.11 |





Table 4 shows the statistical measures between the predicted and actual surface water fraction. Model predicted surface water fraction shows a good agreement with the actual value with an $R^2$ of 0.93 for the training data. When testing using the independent validation dataset, $R^2$ is only slightly smaller, and RMSE and MAE slightly larger, suggesting that the Cubist regression model does not suffer from overfitting. The RMSE and MAE for fluctuating water were slightly larger as compared

to permanent water (Table 4). This suggests that the developed model not only provides accurate results for the static mapping of surface water fraction, but can also be applied effectively for monitoring the dynamics of fluctuating surface water.

## 5.2 Surface water fraction metrics and comparison of products

The mean annual maximum surface water fraction generated from GSW and MODIS-derive product over the 2000-2015 period are displayed in Fig. 4. Overall, the two maps are in good agreement. Visual comparison indicates that our MODIS-derived

product is able to detect narrow rivers with widths covering a couple of MODIS pixels such as the rivers Danube, Euphrates, Po, Rhine, and Tagus. Large differences are evident for some low surface water fraction regions such as parts of Hungary and Ukraine (Fig. 4). These differences likely correspond to the presence of wet meadows, salt marshes and floodplains along large rivers, which are usually saturated and inundated with water during most of the vegetative season (Šefferová Stanová et al., 2008;Stefan et al., 2016). Around the Po River in Italy, where rice paddies are seasonally present, our MODIS product also

shows larger surface water fractions (Fig. 4c). This result suggests that the MODIS-derived surface water fraction has enhanced sensitivity to surface water in wetland areas with emerged vegetation, and this could attributed to the fact that several predictor variables such as LSWI (Xiao et al., 2002) are also sensitive to vegetation water content (Li et al., 2015). Table 5 confirms that the total surface water areas for the Mediterranean calculated from MODIS  is more comparable with the GSW results when only considering areas with higher surface water fraction. For example, when only accounting for pixels with surface

water fractions equal or greater than 50%, the total surface water areas for the Mediterranean based on both data sets are similar (75,107 km$^2$ for GSW versus 73,444 km$^2$ for MODIS). In comparison, only 70,543 km$^2$ water was detected in this region based on the 250 m static water mask from MOD44W. This implies that our MODIS product detects more surface water than other coarser resolution binary maps. The large discrepancy between MODIS surface water fraction and GSW when including low surface water fraction (i.e. threshold =20% and threshold =10%) is probably due to the corresponding mixed pixel effects as

described above and also stated by Klein et al. (2017).



Figure 4 Mean annual maximum surface water fraction maps as determined by (a) JRC's GSW and (b) MODIS time series

surface water fraction for the period 2000-2015, and (c) difference between the two maps. Positive difference values indicate

larger water fraction as detected by MODIS and negative values indicate larger water fraction as detected by JRC's GSW





Table 5 Comparison of total surface water area (km$^2$) as determined from JRC's GSW and MODIS mean annual maximum surface water fraction maps for different thresholds

| Threshold for surface water fraction | Total surface water areas (km$^2$) based on GSW | Total surface water areas (km$^2$) based on MODIS surface water fraction |
| --- | --- | --- |
| 90% | 48,718 | 47,145 |
| 80% | 55,887 | 51,778 |
| 70% | 62,371 | 58,996 |
| 60% | 68,855 | 65,993 |
| 50% | 75,107 | 73,444 |
| 40% | 81,220 | 82,417 |
| 30% | 87,217 | 96,239 |
| 20% | 93,207 | 142,531 |
| 10% | 99,260 | 284,916 |

Figure 5 shows the standard deviation of annual maximum surface water fraction. It indicates that areas of large inter-annual variability agree between MODIS- and GSW-based results. Both indicate a larger variability in surface water fraction in semi-arid and desert climate zones, particularly in the north of Algeria, for the Volga Delta in the Caspian depression, and along the Tigris and Euphrates rivers of Iraq.

Figure 6 displays the seasonality metric for the entire study area, with details for two selected sites shown in panels (a) and (b) of Fig. 7 and Fig. 8. Fuente de Piedra is a seasonally flooded lake which mostly dries out completely in summer (May-September) (Li et al., 2015;Batanero et al., 2017) with the exception of extremely wet years (e.g. 2010, 2011, 2013: Rodriguez-Rodriguez et al., 2016) when water was present throughout the whole year. The differences in water seasonality for Fuente de Piedra between the two products (Fig. 7a, b) can be attributed to the fact that GSW lacks observations in wet seasons, resulting in a reduced water occurrence as compared to our MODIS product. The discontinuities of Landsat record between seasons can affect the accuracy of seasonality information, which was also demonstrated by Klein et al. (2017) and Pekel et al. (2016).
Permanent water is present in parts of Lake Sabkhat al-Jabbul, but the larger part in its center has highly dynamic intermittent water (Fig. 8). Mixed permanent and semi-permanent water are mostly found on the edge of permanent water and in narrow rivers.





Figure 5 Standard deviation of surface water fraction as calculated from (a) JRC's GSW and (b) MODIS annual maximum
surface water fraction maps for the period 2000-2015





Figure 6 Seasonality information derived from time series of (a) JRC's GSW and (b) MODIS surface water fraction for a single year (October 2014 to September 2015)

**5.3 Validation of MODIS time series surface water fraction**

5    Panels (c) of Fig. 7-Fig. 8 show the time series of surface water extent detected by MODIS and GSW for two selected lakes. Fuente de Piedra (Fig. 7c) experiences large temporal variability in surface water extent throughout the year that is well represented by our MODIS product with 464 time steps (Table 6). This variability corresponds closely to the in situ water level data except for extreme wet years (i.e. 2010, 2011 and 2013) when the lake remained flooded through the whole year without



increasing further in size in relation to water level changes (Rodriguez-Rodriguez et al., 2016). GSW has only 72 valid time steps with most observations in dry seasons (i.e. June to October) thus not allowing to appropriately capture the seasonal dynamics, particularly for the November-March period when the lake usually reaches its full surface water extent. Time series water extent of Lake Sabkhat al-Jabbul as determined by our MODIS product had a good agreement with the results from

5    GSW including the seasonal peak extent and the minimum surface water extent during the dry season (Fig. 8c). This implies that the coarse 500 m MODIS data cannot only provide more detailed temporal information (563 MODIS surface water fraction time steps Vs. 69 GSW time steps) (Table 6), but also accurate estimations on surface water area when compared with the results from Landsat 30 m resolution data.

10   Table 6 Number of valid temporal observations for the three lakes based on MODIS surface water fraction and JRC's GSW between February 2000 and October 2015

| Lakes | MODIS surface water fraction | JRC's GSW |
|---|---|---|
| Fuente de Piedra | 464 | 72 |
| Lake Sabkhat al-Jabbul | 563 | 69 |

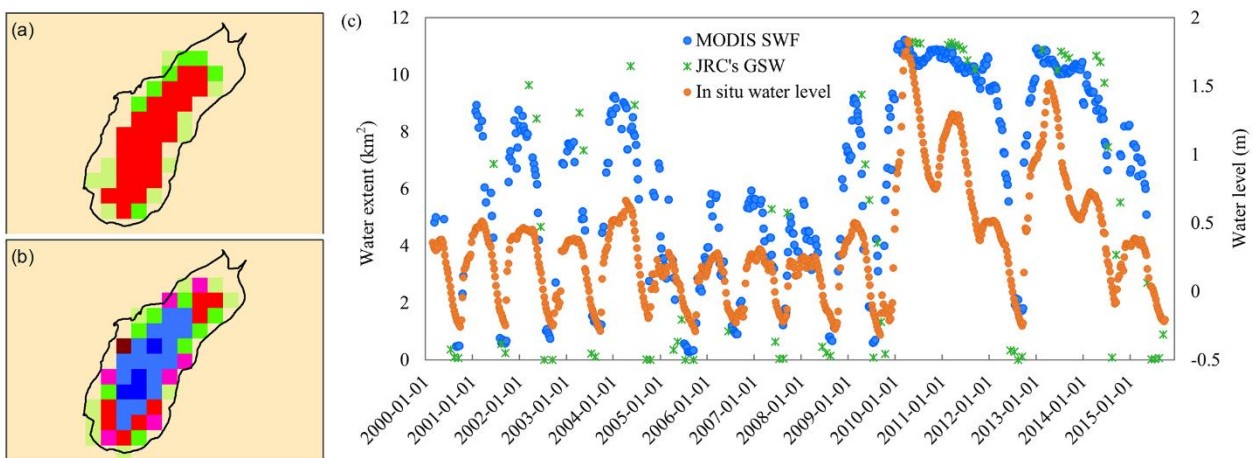

Figure 7  Seasonality information derived from (a) JRC's GSW and (b) MODIS surface water fraction from a single year

15   (October 2014 to September 2015). The colors for (a) and (b) are the same as in Figure 6; (c) Comparison of time series of surface water area (km$^2$) derived from JRC's GSW (showing in green asterisks) with MODIS surface water fraction (blue dots) from 2000 to 2015, along with in situ water level data (orange dots) for Fuente de Piedra, Spain





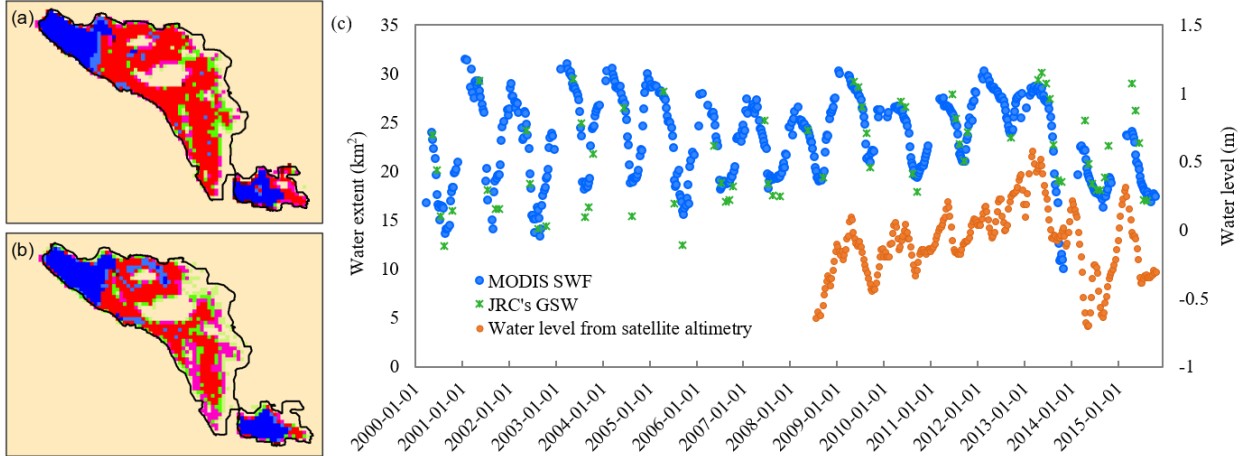

Figure 8  Same graphs as provided in Figure 7, but here showing the results for Lake Sabkhat al-Jabbal, Syria. Water level for this lake is computed from Jason-2/OSTM altimetry and repeats every 10 days

5    Figure 9 compares the MODIS monthly surface water fraction with the GSW monthly water history for Doñana, Spain. The visual comparison shows that the distribution of MODIS surface water fraction had a good agreement with GSW monthly water maps. The 500 m MODIS surface water fraction is able to capture the spatial patterns as detected from the high-resolution Landsat-based GSW dataset. Seasonal drying out and flooding of the wetland is well detected with the MODIS-derived surface water fraction while GSW lacks temporal details for example during large water extents in November and December. MODIS

10    also captured the timing of the maximum water extent (i.e. in January) and water retreat (i.e. in July). This example highlights that the information provided by our MODIS product can contribute to a better understanding of surface water dynamics.



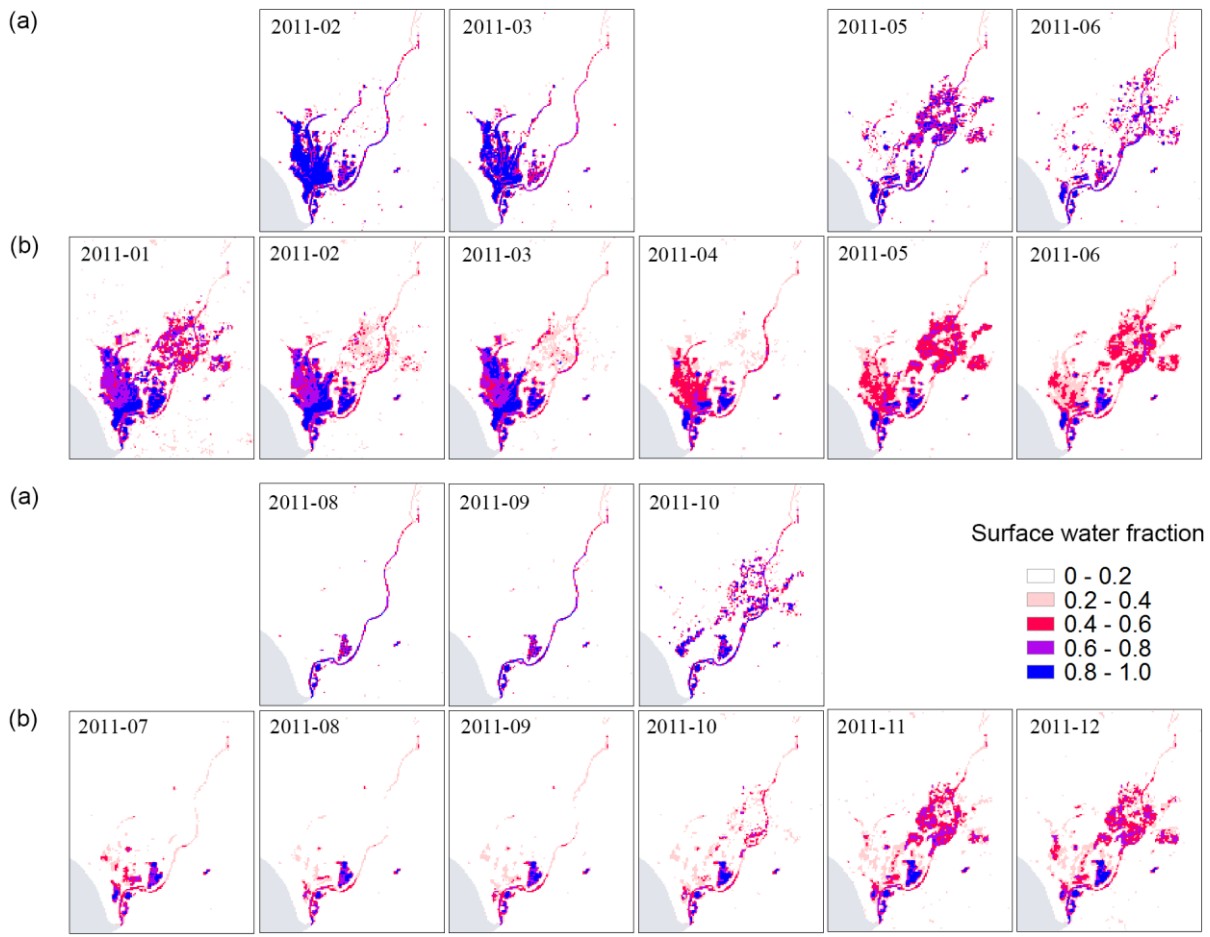

Figure 9  Comparison of monthly water distribution based on (a) Landsat-based GSW dataset and (b) MODIS surface water fraction for Doñana, Spain for 2011

## 6 Discussion

We estimated surface water fraction for the Mediterranean region from MODIS data, and improved on previous efforts to estimate surface water fraction from medium-resolution imagery (MODIS or similar). The prediction accuracy of our model ($R^2 = 0.91$, RMSE = 11.41%, MAE = 6.39%) is higher than the $R^2$ of 0.625 reported by Weiss and Crabtree (2011) who used a linear regression model, and $R^2$ of 0.7 reported by Guerschmann et al. (2011) using a logistic regression model. This research successfully expanded our previous work (Li et al., 2018) by upscaling it from a relatively small region to the whole Mediterranean while retaining a similar high accuracy (both achieved an $R^2$ of 0.91). This attributes to the feasibility and robustness of Cubist regression model to deal with different environmental conditions when training data are collected across a wide geography resulting in varying spectral characteristics.



Secondly, we generated surface water fraction maps at high temporal frequency, which is an advantage over existing fine-resolution datasets. Our MODIS-derived surface water fraction product provides accurately displays spatio-temporal variability of surface water. The comparison with the GSW 30 m product and water level data (Fig.7-Fig. 9) reveal that our

product can efficiently monitor the seasonal, intra-annual and long-term surface water dynamics with good spatial and temporal accuracy. For example, it complements the GSW by allowing for the detection of inundation and recession processes over short time periods and for the better representation of seasonality changes and temporal trends over longer periods. MODIS surface water fraction can also accurately detect the spatial distribution of surface water inclusive of small water bodies (less than one MODIS pixel) and narrow rivers, which are missing in other coarse resolution products using binary classification

method such as MOD44W (Salomon et al., 2004). Metrics derived from the time series MODIS surface water fraction and GSW can reflect different facets of surface water dynamics. The accuracy of some metrics such as seasonality relies on the number of valid observations and temporal interval. Landsat derived metrics might be problematic in areas with large temporal gaps caused by persistent cloud cover, as shown in this study (Fig. 8) and previously by Pekel et al. (2016) and Klein et al. (2017). Even though cloud coverage also limits MODIS observations, the probability of obtaining cloud free observations is

larger (Table 6) due to the daily acquisitions and consequent temporal compositing possibilities. With these advantages, we expect that our MODIS surface water fraction product could fill in important information on surface water for areas and time periods for which cloud-free Landsat acquisition are few or non-existent.

Our MODIS-derived surface water fraction product also has limitations. First of all, it is designed to detect only open surface water, while it may not effectively capture water bodies covered by dense vegetation, such as swamps, lakes with considerable

coverage of aquatic vegetation, and inundated dense forests. Secondly, the MODIS surface water fraction product overestimated small surface water fractions of less than 20% (Table 5), which was also found in previous studies on surface water fraction mapping (Parrens et al., 2017;Li et al., 2018). This overestimation might be attributed to the mixed spectral response of pixels with different land cover types as already demonstrated by many studies (e.g. Guerschmann et al., 2011;Klein et al., 2017). Further work could consider re-assigning those pixels with less than 20% surface water to 0% water

fraction for locations where water is never present according to the GSW maximum water extent. Thirdly, the auxiliary layers utilized for identification of potential areas of commission errors also appear to have some limitations. For example some urban areas might be not mapped in MCD12Q1, and areas of cloud cover and cloud shadow might not be completely removed from the MCD43A4 product. The importance of these limitations may diminish as the quality of these auxiliary layers improves or dynamic datasets rather than static layers are incorporated (e.g. Global Human Settlement Layer: Pesaresi et al., 2016).

Fourthly, the MCD43A4 product also suffers from many missing values, especially in regions with large amounts of precipitation, aerosol concentrations, or snow and ice coverage (Klein et al., 2017). Future research should focus on combining other moderate resolution data (e.g., MOD09) for areas with missing data to ensure a gap-free reconstruction of inland water development for the past and future.





This work can be further scaled up over much larger regions and for shorter (e.g., daily) time intervals. A high-quality training dataset is crucial for the effective application of the rule-base regression model and collection of such training data is time consuming (Sun et al., 2012). In this paper, time series of GSW monthly water history dataset proved to be an efficient basis for building a reliable training dataset. Considering that GSW is globally available, we are confident that our approach can be

scaled to monitor surface water fraction globally with MODIS data. Although the surface water fraction maps were produced with an 8-day time step, the model developed in this paper was actually trained using daily MODIS data and could be directly applied to that temporal resolution. Resulting daily surface water fraction maps could be of high interest for ecological and hydrological research. The recent Global Climate Observing System (GCOS) report states that Essential Climate Variables (ECV) data records requires water extent and lake ice cover with daily temporal resolution and 20 m and 300 m spatial

resolution (Belward, 2016). Our approach makes this requirement for the coarser of the two resolution within reach.

Our MODIS surface water fraction datasets can be beneficial for a large number of applications. It may provide new insights for understanding how climate change, meteorological variability and human activities affect the dynamics of surface water and human livelihoods. The MODIS surface water fraction dataset may also help to improve the calibration and validation of hydrological models, as data on water volume variability are often lacking in areas where in situ measurements are sparse or

inaccessible. Similarly, our long-term records and derived metrics have potential to contribute to the management and conservation of biodiversity and other ecosystem services associated with terrestrial surface water and wetlands.

**7 Conclusion**

We derived an 8-day 500 m resolution surface water fraction product over the Mediterranean for 2000-2017 by applying a global Cubist regression tree model to MODIS and SRTM data. We validated the results with JRC's Landsat-derived GSW

dataset, which resulted in a high overall accuracy ($R^2 = 0.91$, RMSE = 11.41%, MAE = 6.45%). MODIS-derived surface water fraction showed a good spatial and temporal correspondence with JRC's GSW. Comparison with satellite altimetry and in situ water level data for selected lakes demonstrated the ability of MODIS surface water fraction to effectively monitor seasonal and inter-annual changes in surface water extent. Our dataset provides a consistent and long-term record (18 years) of 8-day water fraction dynamics for the Mediterranean region, and complements fine spatial resolution surface water products,

especially in regions where such products have long temporal and spatial data gaps due to both the limited number of acquisitions and persistent cloud cover. Our approach is also promising for monitoring surface water fraction at global scale and at daily interval.

**Data availability.** The final derived 18 years of surface water fraction maps for the Mediterranean region can be access through

https://surfdrive.surf.nl/files/index.php/s/ebDQ7FNvCTeAziQ (password: WaterFraction18), and will be open available publicly through https://doi.org/10.17026/dans-xrz-y92s at the time of publication.




**Author contributions.** LL and VA designed the experiment. LL performed all analysis and developed the 18-year database. LL prepared the manuscript with contributions from all co-authors. All authors have approved the final version.

## Acknowledgements

This work of the first author was supported by the China Scholarship Council under Grant 201206180040 and co-funded by ITC Research Fund. We would like to thank Alan Belward and Jean-François Pekel (Joint Research Centre) for their support in accessing the global surface water datasets. Many thanks to the authorities of the Fuente de Piedra Natural Reserve and the Junta de Andalucía for providing the in situ water level data used in this study. We also thank Willem Nieuwenhuis for his technical support.

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
