# Peer review of "A New Dense 18-Year Time Series of Surface Water Fraction Estimates from MODIS for the Mediterranean Region"

_Hydrology and Earth System Sciences, 2019_

## Referee Comment (RC1) · Anonymous Referee #1 · 5 Mar 2019

Dear authors,

I enjoyed reading your manuscript on the use of MODIS data to develop and validate the surface water fraction estimates for the Mediterranean Region. Eventhough the surface water derivation from EO data was studied a lot in the last years, the authors demonstrate the importance of further refinement of knowledge on the surface water dynamics.

The temporal frequency of the Landsat dataset used to generate the GSW dataset (for the years before 2013) is still insufficient to represent the actual surface water dynamics, especially for the areas where the surface water is rapidly changing (reservoirs,

floodplains, dynamic coastal areas). I liked how Figure 2 was used to demonstrate the missing monthly data history over time. However, I've missed the reference to that figure in the text, and the explanation on how this was determined? Does the figure show the missing value pixels (masked out) over sample locations shown in Figure 1? I'd even suggest that all figures should be referenced somewhere in the text.

P7L11 Please include a proper citation for GEE, e.g. https://www.sciencedirect.com/science/article/pii/S0034425717302900

Did you consider making your GEE scripts public, under a proper license? This will enable the reproducibility of your research and should be trivial to do.

P14 Validation: Spain is a very sunny area, how does your algorithm perform in temperate areas?

Figures 4-6 - you may need to upscale these map (using reduceRegolution() in GEE), it's really hard to distinguish waterbodies at that scale, making these figures more or less useless.

Figures 7-8 - add a scatterplot for both charts will significantly improve understanding of the differences between these datasets

Sincerely yours, Referee

―――――――――――――――――

---

## Referee Comment (RC2) · Anonymous Referee #2 · 14 Apr 2019

This well-organized manuscript presents a valuable data set and an interesting method to monitor surface water fraction/extent variation in large spatial scale and high temporal resolution. The combination of a medium resolution MODIS surface reflectance product and a well-tuned Cubist regression model provided a sub-pixel estimation of water fraction with satisfactory performance compared with GSW and previous MODIS product on surface water. Apparently, the authors have made their efforts to further improve the quality of the data set by using ancillary data such as DEM and Land Cover to eliminate possible contaminated pixels. Besides the cross validation with GSW, the authors also compared the generated data set with altimetry data in certain lakes. Given the importance of high frequency monitoring of surface water in water management, it

is positive that this work will benefit the scientific community as well as public decision makers.

In revising the manuscript, following issues were encountered and I suggest authors to provide further explanation:

1) In Page 11, Table 2, the author listed the input predictors for Cubist regression model. However, it seems to me that TCWI and TCBI might be redundant variables because they are linear combination of MODIS individual bands and the regression model is also linear. Have you ever tried to remove these two predictors to simplify the model input? If so, how did it work?

2) In Page 17, Table 5 listed the comparison results between water extent determined from GSW and MODIS at different thresholds. Though the author mentioned that the generated MODIS surface water product tends to overestimate the water extend due to mixed pixel effects at low water fraction thresholds, it's still confusing that the difference between the two products (MODIS and GSW) doesn't decrease with the increasing threshold higher than 40% or 50%. In fact, the generated MODIS product tends to underestimate the water extent compared with GSW when using larger thresholds.

3) In Page 20, figure 7 (c) and Page 21, figure 8 (c), the altimetry water levels and MODIS generated water areas were compared. However, it could be more convincing to compare two time series with same physical meaning. The altimetry water levels can be transformed into water areas using hypsometric curves (some can be found in existing data sets such as Hydroweb), vice versa. By doing this you can calculate some metrics to better describe the agreement of two data sources.

4) The potential of the data set in water management could be better illustrated. High temporal resolution surface water areas can benefit some studies that use water area information as input for hydrological modeling in ungauged basin, e.g., Huang et al. (2018) used river widths generated from Landsat and Sentinel-2 to calibrate the parameters of a distributed hydrology model in Upper Brahmaputra River.

Reference

Huang, Q., Long, D., Du, M., Zeng, C., Qiao, G., Li, X., Hou, A., and Hong, Y.: Discharge estimation in high-mountain regions with improved methods using multisource remote sensing: A case study of the Upper Brahmaputra River, Remote Sensing of Environment, 219, 115-134, 2018.

---

## Author Comment (AC2) · 9 Jun 2019

The authors thank the reviewer for his/her useful comments, which will help improve the manuscript. Please see the attached supplement for responses to the reviewer's comments.

Please also note the supplement to this comment: https://www.hydrol-earth-syst-sci-discuss.net/hess-2019-5/hess-2019-5-AC2-supplement.zip

---

## Author Comment (AC3) · 9 Jun 2019

Please see attached the revised paper with track changes.

Please also note the supplement to this comment:
https://www.hydrol-earth-syst-sci-discuss.net/hess-2019-5/hess-2019-5-AC3-supplement.pdf
* * *

---

## Author Response (AR1)

**Response to Reviewer #1**

**General comments:** I enjoyed reading your manuscript on the use of MODIS data to develop and validate the surface water fraction estimates for the Mediterranean Region. Even though the surface water derivation from EO data was studied a lot in the last years, the authors demonstrate the importance of further refinement of knowledge on the surface water dynamics.

We thank the reviewer for her/his kind words about our study and valuable suggestions to improve the quality of the manuscript. Below we explain in detail how we have incorporated these comments in the manuscript.

(1) **Comment:** The temporal frequency of the Landsat dataset used to generate the GSW dataset (for the years before 2013) is still insufficient to represent the actual surface water dynamics, especially for the areas where the surface water is rapidly changing (reservoirs, floodplains, dynamic coastal areas). I liked how Figure 2 was used to demonstrate the missing monthly data history over time. However, I've missed the reference to that figure in the text, and the explanation on how this was determined? Does the figure show the missing value pixels (masked out) over sample locations shown in Figure 1? I'd even suggest that all figures should be referenced somewhere in the text.

**Response:** The reviewer is correct that Figure 2 indeed demonstrates the proportion of missing data/valid data in the Landsat-based GSW monthly dataset over time. Figure 2 actually shows the percentage of pixels with valid observation calculated over the entire Mediterranean region shown in Figure 1. We realized that this was not well explained in the text. We consequently revised the text and the caption of Figure 2 to make it clearer.

**Changes in manuscript:**

P4L12: added "the percentage of pixels with valid observation in JRC's GSW monthly water history dataset for each month between January 2000 and October 2015 calculated over the entire Mediterranean area. The figure illustrates".

P4L15: added "Mediterranean".

P5L3: replaced "valid data points" with "pixels with valid observation".

P5L4: added "entire".

(2) **Comment:** P7L11 Please include a proper citation for GEE, e.g. https://www.sciencedirect.com/science/article/pii/S0034425717302900

**Response:** We have included the suggested paper in our citation.

**Changes in manuscript:** P7L11: added a reference "Gorelick et al. 2017".

(3) **Comment:** Did you consider making your GEE scripts public, under a proper license? This will enable the reproducibility of your research and should be trivial to do.

**Response:** This is a good suggestion. At present, the work of this paper was partly done in GEE (i.e. collecting training and validation dataset from GSW monthly history data) and partly in the R software environment (i.e. building the regression model and applying the model to MODIS time series data). In our opinion the code at present still lacks appropriate documentation to make it publicly available. We are nonetheless open to share the code with interested researchers, and now mention this in the data availability section (P27).

**Changes in manuscript:** P27L8: the data availability section now reads "The final derived 18 years of surface water fraction maps for the Mediterranean region are open available through https://doi.org/10.17026/dans-xrz-y92s. The GEE and R code used for this paper are available on request from the first author."

(4) **Comment:** P14 Validation: Spain is a very sunny area, how does your algorithm perform in temperate areas?

**Response:** We are currently finalizing a follow-up manuscript, which we expect to submit soon. In that manuscript we provide an in-depth evaluation of this dataset for monitoring surface water dynamics across various climate regions. We did realize however that the title of the manuscript, as well as the titles of Sections 4.3 and 5.3 were suggesting a more formal validation effort; as a consequence we have adapted those. For the present paper, we think that the examples can illustrate the potential of our dataset, which is the main purpose here. Nonetheless, we can share with the reviewer an example for a temperate areas (taken from our manuscript under preparation). The figure below shows the evaluation results for Bardaca Wetland, a Ramsar wetland in Bosnia and Herzegovina. Monthly water extent time series obtained from MODIS SWF show a good agreement with those from JRC'GSW ($r$=0.88).

[Figure]

Figure 1 Comparison of monthly water extent (km$^2$) time series obtained from MODIS SWF and JRC'GSW.

**Changes in manuscript:** P1L1: replaced "Development and Validation of a" with "A New".

P14L1: the sub-title of Sections 4.3 now reads "Demonstrating the representation of surface water dynamics by the new MODIS dataset".

P20L4: the sub-title of Sections 5.3 now reads "MODIS-derived surface water dynamics for selected lakes".

(5) **Comment:** Figures 4-6 - you may need to upscale these map (using reduceRegolution() in GEE), it's really hard to distinguish waterbodies at that scale, making these figures more or less useless.

**Response:** We agree with the reviewer that interpreting a 500 m resolution map for a large region is cumbersome. However, we want to show results for the entire region, and given that surface water presence can be rather local. We are in favor of keeping the maps in the way that they are, because in our view main features and differences are visible. Note that in the case of Landsat-based GSW, we have indeed aggregated the Landsat 30 m pixels to 500 m pixels. We do realize however that the resolution of the figures was not optimal in our initial submission. We have now provided the same Figures 4-6 with higher quality and larger size, which can be downloaded by readers from the full-text HTML version online. In these improved figures, individual MODIS grid cells can be clearly separated.

**Changes in manuscript:** P17L1: replaced "determined by" with "obtained from"

P17L3: added "that our MODIS dataset detected a"

P17L3: deleted "as detected by MODIS and negative values indicate larger water fraction as detected by"

**(6) Comment:** Figures 7-8 - add a scatterplot for both charts will significantly improve understanding of the differences between these datasets

**Response:** Following the reviewer's suggestion, we have added two scatterplots for Figure 7 and 8 in the revision.

**Changes in manuscript:** P22-23: updated Figure 7-8.

P20L5: replaced "c" with "d"

P20L6: replaced "7c" with "7d"

P20L7: replaced "464" with "461"

P20L8: added "($\rho$ =0.95)."; replaced "except for" with "Note that in"; deleted "when"

P21L1: replaced "through" with "throughout"; deleted "whole"

P21L2: added "Water extent derived from MODIS SWF also had a close match with that from GSW (Figure 7c; $r$=0.95)."

P21L3: replaced "72" with "73"

P21L5: added "showed relative high correlation with water level data ($\rho$=0.69). It also"

P21L6: replaced "results" with "water extent derived"

P21L7: added "and d; $r$=0.88"

P21L11: In table 6 replaced "464" with "461"; replaced "72" with "73"; added one row "Doñana 714 70"

P22L4: added "Scatterplot of water area obtained from JRC's GSW versus that from MODIS SWF. $r$ represents the Pearson's correlation between two datasets; (d)".

P22L8: added ". $\rho$ represents the Spearman rank correlation between water area and water level"

**Response to Referee #2**

**General comments:** This well-organized manuscript presents a valuable data set and an interesting method to monitor surface water fraction/extent variation in large spatial scale and high temporal resolution. The combination of a medium resolution MODIS surface reflectance product and a well-tuned Cubist regression model provided a sub-pixel estimation of water fraction with satisfactory performance compared with GSW and previous MODIS product on surface water. Apparently, the authors have made their efforts to further improve the quality of the data set by using ancillary data such as DEM and Land Cover to eliminate possible contaminated pixels. Besides the cross validation with GSW, the authors also compared the generated data set with altimetry data in certain lakes. Given the importance of high frequency monitoring of surface water in water management, it is positive that this work will benefit the scientific community as well as public decision makers.

In revising the manuscript, following issues were encountered and I suggest authors to provide further explanation:

We thank the reviewer for her/his positive feedbacks and valuable comments and suggestions to improve the quality of the manuscript. Please find below our responses and how we have incorporated these comments in the manuscript.

 (1) **Comment:** In Page 11, Table 2, the author listed the input predictors for Cubist regression model. However, it seems to me that TCWI and TCBI might be redundant variables because they are linear combination of MODIS individual bands and the regression model is also linear. Have you ever tried to remove these two predictors to simplify the model input? If so, how did it work?

**Response:** The predictor variables are used in two ways in the Cubist regression model. Firstly, they set rule conditions to split the samples into smaller subsets. Secondly, they are used to build linear regression models related to the rule conditions. The reviewer's concern only relates to the latter purpose, and in this regard, MODIS individual bands can indeed replace TCWI and TCBI to build linear regression models. However, TCWI and TCBI, especially their temporal characteristics, are important variables to set rule conditions. This was demonstrated in our previous paper (Li et al. 2018), in which we measured the relative importance of all variables for estimating surface water fraction in two small regions based on its usage in the rule conditions and in the linear regression models. The results (please see the figure below) showed that the temporal characteristics of TCBI and TCWI such as the annual max/min/mean are frequently used for setting rule conditions following TWI (Topographic Wetness Index) and NIR. In addition, our previous paper also showed that models using all variables (inclusive of temporal variables) achieved higher prediction accuracies as compared to simpler models (exclusive of temporal variables). Therefore, we conclude that TCBI and TCWI are not redundant but instead are important variables in the

Cubist regression model. We realized that the text did not specify this, but have now revised in section 4.1.3 to clarify this.

[Figure]

Figure 1. Twenty predictor variables with the highest relative importance for estimation of surface water fraction. Importance is measured as variable usage (%) in the rule conditions (b) and in the linear models (c) with the Cubist model (Li et al. 2018)

Reference: Li, L., Vrieling, A., Skidmore, A., Wang, T., & Turak, E. (2018). Monitoring the dynamics of surface water fraction from MODIS time series in a Mediterranean environment. *International Journal of Applied Earth Observation and Geoinformation*, 66, 135-145.

**Changes in manuscript:**

P11L1: added "also" and ". These temporal summaries were demonstrated to be an important input"

P11L2: added "in our previous study (Li et al. 2018)."

2) **Comment:** In Page 17, Table 5 listed the comparison results between water extent determined from GSW and MODIS at different thresholds. Though the author mentioned that the generated MODIS surface water product tends to overestimate the water extend due to mixed pixel effects at low water fraction thresholds, it's still confusing that the difference between the two products (MODIS and GSW) doesn't decrease with the increasing threshold higher than 40% or 50%. In fact, the generated MODIS product tends to underestimate the water extent compared with GSW when using larger thresholds.

**Response:** We thank the reviewer for raising this point. In fact, machine learning approaches such as Cubist regression model and random forest often overestimate small values and underestimate large

values when estimating fractional cover of land surface components (e.g. Huang et al., 2014; Li et al. 2018; Wang et al., 2017). As a consequence, our MODIS surface water product tends to overestimate water extent at low water fraction and underestimate water extent at high water fraction. In the previous version of the manuscript, we only discussed the overestimation but not the underestimation. We now have added the reasons why large water fraction covers were underestimated by our MODIS product.

**Changes in manuscript:**

P16 L26, replaced "The large discrepancy between MODIS surface water fraction and GSW when including low surface water fraction (i.e. threshold =20% and threshold =10%) is probably due to the corresponding mixed pixel effects as described above and also stated by Klein et al. (2017)." with "Nonetheless, our MODIS product detects less surface water compared to GSW for larger thresholds (≥50%), whereas it detects much more surface water than GSW for small thresholds (≤20%). This confirms an earlier finding that machine learning approaches such as Cubist and random forest often underestimate large values and overestimate small values when estimating fractional cover of land surface (e.g. Huang et al. 2014; Li et al. 2018; Wang et al. 2017). In addition to effects of mixed pixels (Klein et al. 2017), the most obvious reason is because regression techniques used in such approaches fit linear equations to relationships that may not be linear over the entire range of values."

3) **Comment:** In Page 20, figure 7 (c) and Page 21, figure 8 (c), the altimetry water levels and MODIS generated water areas were compared. However, it could be more convincing to compare two time series with same physical meaning. The altimetry water levels can be transformed into water areas using hypsometric curves (some can be found in existing data sets such as Hydroweb), vice versa. By doing this you can calculate some metrics to better describe the agreement of two data sources.

**Response:** We agree with the reviewer that it would be preferable to compare water area with water area, and that hypsometric curves are a way to transform our in-situ water level data in water area. Unfortunately, such curves are to the best of our knowledge not presently available for the lakes presented here. Although arguably these could be constructed using remote sensing data, we prefer to do a direct comparison of our water area estimates with a directly-measured in-situ quantity, i.e. water level, despite that we acknowledge their different physical meanings. Our illustration is mainly intended to demonstrate that the temporal behavior of our estimates with water level correspond, which is to be expected. Rather than a normal regression analysis, a Spearman rank correlation is a better way to assess their relationship. We have now calculated the Spearman rank correlation between water level and water area derived from our MODIS product and JRC's GSW, and added the results in Figure 7 (d) and 8 (d).

**Changes in manuscript:** P21-22: updated Figure 7-8.

P14L15: added "We calculated the Spearman rank correlation (ρ) between water level and water area derived from MODIS SWF and JRC's GSW data to assess the correspondence between these datasets."

P22L8: added "ρ represents the Spearman rank correlation between water level and water area".

4) **Comment:** The potential of the data set in water management could be better illustrated. High temporal resolution surface water areas can benefit some studies that use water area information as input for hydrological modeling in ungauged basin, e.g., Huang et al. (2018) used river widths generated from Landsat and Sentinel-2 to calibrate the parameters of a distributed hydrology model in Upper Brahmaputra River.

Reference: Huang, Q., Long, D., Du, M., Zeng, C., Qiao, G., Li, X., Hou, A., and Hong, Y.: Discharge estimation in high-mountain regions with improved methods using multisource remote sensing: A case study of the Upper Brahmaputra River, Remote Sensing of Environment, 219, 115-134, 2018

**Response:** We have incorporated this suggestion and added more content in the Discussion section to better illustrate the potential and application of this dataset. We have also added the suggested paper.

**Changes in manuscript:** P26L14: added "For example, it could be used as a monitoring tool for analyzing hydrologic extremes such as floods and droughts, detecting abnormal changes of wetland hydrology, capturing short-duration events, identifying newly-formed and disappearing water bodies, and estimating global water loss."

P26L14: replaced "can be beneficial for" with "may benefit".

P26L18: added "For example, the water area can help to estimate a series of hydrological parameters such as water discharge (Huang et al. 2018) and water volume (Busker et al. 2019; Cael et al. 2017; Duan and Bastiaanssen 2013; Tong et al. 2016). This would be particularly useful for areas where in situ measurements are sparse or inaccessible."

P26L21: replaced "It may provide new insights" with "Closely monitoring hydrological variability is important"

[revised manuscript text omitted]